# From ice cores to dinosaurs: physical collections managers' research data curation perceptions and behaviors

Bradley Wade Bishop[1]*, Jaxx Fox[1], Sidney Wanda Taylor Gavel[1], Emily Grace Chapin[1], Sarah Kansa[2]

**1** School of Information Sciences, University of Tennessee, Knoxville, Tennessee, United States of America, **2** Archaeological Research Facility, University of California, Berkeley, California, United States of America

* wade.bishop@utk.edu

**Data availability statement:** Bishop BW. Collections Discovery and Curation Behavior. Ann Arbor, MI: Interuniversity Consortium for

## Abstract

Physical collections provide the tangible objects that when analyzed become data informing all sciences. Physical collection managers aim to make physical objects discoverable, accessible, and reusable. The volume and variety of physical collections acquired, described, and stored across decades, and in some cases centuries, results from large public and private investments. The purpose of this study is to understand the curation perceptions and behaviors of physical collection managers across domains to inform cross-disciplinary research data management. Ten focus groups were conducted with thirty-two participants across several physical collection communities. Participants responded to open-ended questions about the data lifecycle of their physical objects. Results indicated that physical collections attempt to use universal metadata and data storage standards to increase discoverability, but interdisciplinary physical collections and derived data reuse require more investments to increase reusability of these invaluable items. This study concludes with a domain-agnostic discussion of the results to inform investment in cyberinfrastructure tools and services.

## Introduction

Museums, academic institutions, federal agencies, and individual researchers keep a plethora of samples and objects for research purposes across domains. The data derived from these physical collections informs scientific discovery, but often aggregating data within even a single domain relies on navigating institutional and discipline-specific repositories. Differing curation practices, shifts in data collection methodology, and changing theoretical and funding priorities make the US physical collection infrastructure a patchwork quilt of objects and associated metadata and derived data from a world of objects of all sizes and forms. Any exact science builds knowledge from these invaluable collections collected once in real-time without substitute. In today's context of AI readiness and public access, building a cyberinfrastructure atop this foundation should include revisiting the existing practices in managing physical collections and subsequent research data curation processes. Scientific domains

Political and Social Research [distributor]; April 25, 2024. https://www.openicpsr.org/openicpsr/project/201362/version/V1/view

**Funding:** Institute of Museum and Library Services' (IMLS) Laura Bush 21st Century Librarian grant program (RE-13-19-0027-19). Collaborative Analysis Liaison Librarianship. (2019-2024). The funders had no role in the study design, data collection and analysis, decision to publish, or preparation of the manuscript.

**Competing interests:** The authors have declared that no competing interests exist.

that rely on physical collections to create knowledge work in today's machine-actionable context. The need for more findable, accessible, interoperable, and reusable (FAIR) data, that also accounts for ethical reuse (i.e., CARE Principles for Indigenous Data Governance), as well as funders' expectations for data as open as possible [1–3]. This study helps understand the curation perceptions and behaviors of physical collections managers. Ten focus groups were conducted across scientific communities that build understandings from physical collections and samples, including anthropology, archaeology, paleontology, ichthyology, herbaria, entomology, herpetology, and geology. Current perspectives on curation and behaviors in practice could distill the most important reuse facets that could inform the design of the cyberinfrastructure tools and services to improve reusability of all physical collections and resulting data.

## Literature review

Physical collection managers' responsibilities include all the preservation and storage efforts to save items that represent life and the abiotic world in-perpetuity. Preservation and storage are inherently necessary for reuse. As tools for empirical measurements advance and the paradigms in domains shift the research questions asked of these objects, the scientific value of physical collections fluctuates. Aggregated data stemming from the physical representations give scholars the potential to ask questions at different time-scales across the planet. There may be a lot to unpack from just one object, which is the case for Antarctic ice cores that capture the atmosphere and DNA of microscopic species from thousands of years ago. The potential undiscovered knowledge for biology and geology waiting in a drawer of fossilized brachiopods must first be indexed prior to addressing any evolutionary or geologic research questions. Museums and similar entities by other names hold all types of things gathered prior to today's taxonomies, ontologies, and locality precision techniques that may have more to offer, especially when one considers every rock shaving, bone fragment, fish floating in formalin, or ancient tool is a singular instance of a thing. How items are curated informs all potential reuse and summarizing the perspectives and behaviors of collection managers across domains gives a framework for the discussion of the facets to include in the design of the cyberinfrastructure for all physical specimens.

Physical specimens could be anything from former living organisms to geological samples to human cultural objects. Abiotic items have different ethical and legal considerations compared to living organisms or human cultural objects. The broadest legal frameworks these items fall under are the UNESCO 1970 Convention for cultural heritage objects which includes human remains, cultural objects, and fossils [4], and the Convention on Biological Diversity which broadly protects biological diversity as a whole as well as sustainable use and equitable benefits of use [5], both of which are international agreements through the United Nations. Unfortunately, none of these international frameworks have legal action unto themselves and require domestic laws within each country to enforce compliance with these agreements [6]. This leads to a hodgepodge of enforcement [7,8]. As technology progresses and biodiversity is threatened, the importance of these physical specimens only increases *in situ* and in collections [9–11]. Who owns what property and what rights these objects may have should not be overlooked as a move to AI readiness and more open science results in increased reuse, including potential misuses.

Regardless of intellectual property considerations, each data story of physical objects varies across disciplines. The preservation and management strategies over time rely on the expertise of specialists with unique training to describe, preserve, and make available various things.

In today's data-intensive paradigm in sciences, Research Data Management (RDM) provides another layer of considerations for managers beyond the efforts to make the underlying physical collections accessible and reusable. The scans, images, documentation, and other materials provide digital representations into these items wherever they may be situated. RDM entails appraising, preserving, storing, and transforming digital objects to enable reuse beyond the original data collector. Increasingly research institutions and funding agencies have both incentivized and mandated data sharing as a way of supporting a more open and data-intensive research enterprise. As FAIR-aligned data becomes central to scientific advancements, the need to understand how humans perceive and act with physical collections and data will become increasingly important as these procedures undergird all downstream reuse. For the physical collection manager, their default data originates from captured tangible items and their associated ancillary metadata. Associated data for physical objects may include derivatives that when aggregated, analyzed, and transformed, lead to new reuses. These derivatives may include a description of the object (e.g., metadata), provenance documentation, imagery, field notes, other data of many types (e.g., DNA), and even CT and 3D scans in rare instances.

Across science an increasing level of scrutiny on reproducibility has accelerated a move toward open data access [12–15]. New journal requirements for provision of supporting data [16,17]; and also data management demand have led to more effective data discovery, access, reuse, and long-term preservation (e.g., Data Management and Sharing Plans). These movements align with long-standing ethical principles in scientific disciplines concerning public ownership, stewardship, public education, and conservation of the objects that fuel knowledge creation. These new pressures put more emphasis on making all items as findable, accessible, interoperable, or reusable as possible.

Physical data managers faced with concerns related to tangible objects also must gain skills to foster reuse of resultant data. Digital objects have their own data curation challenges related to potential hardware failures, media degradation, software obsolescence, and inadequate documentation. Much of what is not yet lost, is not discoverable or accessible and researchers warn that the knowledge held in these objects will ultimately be lost unless data governance practices in all RDM settings change dramatically [18–20]. First, researchers have concerns about data integrity, specifically, data quality, inadequate documentation and subjectivity in determining what data is worth sharing. Second, there are concerns in each of the qualitative studies cited about maintaining responsible conduct in research. Researchers expressed concern about the misuse of data, protecting intellectual property, privacy/confidentiality issues, and ways to maintain control over the objects and data while allowing reuse. Third, researchers expressed feasibility challenges in sharing data, including the lack of infrastructure support, the time and effort required, and skill deficits in data for sharing. Finally, a dearth of incentives for resharing data—foremost additional funding. This current study provides insights into perceptions and behaviors of physical collection managers across several disciplines reliant on the reuse of physical collections and provides a domain-agnostic reuse facets matrix to inform investment in cyberinfrastructure tools and services.

## Methods

The study uses the data curation profiling (DCP) method to capture all actions across the data lifecycle [21,22]. Participants responded to these open-ended questions that relate to aspects of the data lifecycle–the accession, organization, storage, and use of their collections. All questions were pilot tested with two collection managers and revised accordingly.

### Collection overview

1. Approximately, what is the size of the collection? This includes both the typical size of the items in the collection, and the size/scope of the collection as a whole.
2. What is the average size of the collection's objects? (units of measurement, this may be an individual object, or the size of an entire collection depending on the scope of use.)
3. How is your collection typically stored (e.g., what digital format/type of physical storage. May address both.)?

### Accession and organization

4. How was the collection originally collected/discovered (i.e., approaches/tools/software)?
5. How was the locality of the collection samples determined? Place/time?
6. How did you or your organization acquire or gain access to the collection?
7. Describe briefly the way the collection is currently organized (e.g., sample name conventions, existing metadata)?
8. Does your collection have a persistent identifier (e.g., doi)? What type of identifier do you use?
9. Is your collection findable (i.e., indexed, online catalog)?

### Storage

10. Where are the collections currently stored (both physically and digitally)
11. Are there other representations of the collection (e.g., 3D model, photos)?
12. Who is primarily responsible for managing the collection?

### Use

13. Who are the current and potential users of this collection?
14. What are the main uses of this collection? What additional potential uses?
15. Does the collection have any use constraints?
16. Does the metadata provide sufficient information for use?
17. Are you aware of any other issues about the collection that may impact use?

Purposive recruitment occurred through personal contacts from related projects and organizations that focus on understanding and improving discovery, access, use, and curation of physical research objects across several domains. A total of thirty-two participants from several physical sample communities participated in ten focus groups (https://isamplesorg. github.io/home/). Snowball recruitment led to participant across the following disciplines, including anthropology, archaeology, botany, geology, ichthyology, entomology, herpetology, and paleontology. The participants were not part of the research team, but all full-time physical collection managers working at institutions housing and making access possible for other reusers.

All focus groups occurred from December 2022 to October 2023 to reach an interdisciplinary mix of the types of curation activities across several types of different physical objects. The meetings were recorded in Zoom with closed captioning. Additional transcription occurred prior to coding. The focus groups had an average of three participants each, two with four and one with two. Focus group recordings were coded in NVivo. Open coding was used to capture perceptions and actions conducted in managing these physical collections.

Coders revisited the codebook with each additional community to ensure consistent use of terms across similar concepts.

## Results

The following presents results using codes grouped into sections related to collection overview, storage, data collection, metadata, organization, findability, and use. Relevant responses to different parts of the data lifecycle appeared throughout the focus groups. Although the DCP questions are linear, several participants mentioned items in the collection overview that related to later questions about accession, organization, storage, and use. In fact, each focus group and the mix of participants altered which aspects were focused on, but the question order was consistent. Therefore, the results are presented according to the broad sections of the codebook–*Collection Overview; Storage; Data Collection; Metadata; Accession; Findability; Reuse/Use*.

### Collection overview

The volume and variety of physical collections was best stated by a paleontology participant "we cover the entire fossil record from micro fossils which are microscopic in scale all the way up to dinosaurs and whales" (FG3-C). The collections scope mirrors the complexity of each research area and includes the codes and counts–*collection size* (41), *collection space* (10), *item size* (40), and *previously living* (110) and *non-living* (13) items. Codes to differentiate between types included *previously living* and *non-living* items. The previously living organisms like whole or partial skeletons, whole organisms preserved in jars (i.e., "pickled critters"), hard tissue (e.g., teeth, coral), soft tissue, blood, DNA samples, hair, and so forth. The named organisms include humans, other mammals, reptiles, fish, insects, vascular plants, fungi, algae, and dinosaurs. The abiotic, or *non-living* code, items discussed in the focus groups included all rocks, cores, seismic data, ceramic shards, figurines, the other items mentioned include photographs, 3D models, and books.

The potential *collection size* largely depends on the type of objects. For example, fungi take up less space than human skeletal remains. Responses to this question depend on what might be counted and how items are counted. Ichthyology collections had approximately 15,000 lots with everything from "small tadpoles and eggs all the way up to fully taxidermied anacondas or crocodilians, or leatherback and loggerhead sea turtles" (FG10-C). Whereas entomological collections had up to an estimated guess of 30 million specimens. For geologic core collections a total length in meters is commonplace, but other geological items may have countless shavings and cuttings across millions of boxes and quantifying is more of a challenge as it is a moving target that is also breaking into pieces. Also, across collections similar items are grouped in boxes, buckets, and bags that make an exact count challenging (e.g., a bag labeled *animal bones*). In herbaria, like all life there are a great many things in dazzling variety. As FG7-A put it "we have a large fruit and cone collection that we're currently rehousing into boxes, and those are all crazy sizes." Collection size is key to estimating budget and storage considerations in digital curation, but in these focus groups accurately estimating precise sizes was a challenge. Cloud storage can be a substantial and reoccurring expense and all the aspects of storing and protecting physical collections (i.e., Integrated Pest Management) account for considerable costs. Focus groups also discussed the size of the average physical item, which not surprisingly participants struggled to answer due to the wide range of sizes. Herbaria do have standard sheets to store specimens in folders and entomological collections use certain-sized boxes. In the ichthyology focus group, the diversity of fish was best summed up with this statement "most of our jars are 4 to 8 ounces [for] smaller fishes, all the way up to

an arapaima that we had cut in half to put into an 80-gallon tank" (FG5-D). Still, a diversity of physical items in these collections was expected. Storage considerations and costs multiply to store these tangible and digital objects as access and use of these items increases.

An additional code emerged related to *collection space* to acknowledge this pressing issue for all collections. Collection managers must maximize existing space by adding compact shelving units and one manager had plans to start stacking tanks vertically. Biocollections kept in flammable liquids, such as wet specimens, must meet local fire regulations including firewalls, specific HVAC systems, sprinklers, and fire suppression systems. Those stipulations also limit where items can be safely kept. For both tangible and digital objects, size continues to be a moving target that makes space either physical or virtual another ever increasing cost. FG2-C "our donated skeletal collection [...] has more than 1,900 individuals in it. I can't give you an exact number, because that changes every week." If this is the case for the physical collections with human rights, one can imagine how more challenging the tracking of smaller species and shards of non-living items can be. Compared to human remains, an entomology collection manager faced the reality that "we have a lot of specimens that might not fit coming up soon" (FG9-B). Fortunately, the remainder of the DCP questionnaire translates more easily to physical collections.

## Storage

Due to the variety of specimens and other tangible objects, the physical storage differed across collections, with similarities within types. The storage codes and counts include *physical storage* (66); *lots* (9); *digital storage* (70); and *file* or *data types and formats* (33). Physical storage included a range of container sizes and materials. Bones and other dry items were kept in cardboard boxes, plastic bags, and other plastic containers. Archaeological collections are often sorted into zipped plastic bags before being stored in appropriately sized boxes with lids of either cardboard or steel. This is quite similar to anthropology collections which are stored in standard size bankers boxes. Insects were kept either in boxes, pinned, or loose in envelopes. Plant specimens were mounted on herbarium sheets. Ichthyology collections house their ethanol- and alcohol-preserved specimens in stainless steel tanks, glass jars of various sizes, and 5-gallon plastic buckets. Specific to ichthyology is the term *lots*––a jar that may hold a group of individual organisms that are either the same species or related taxonomically or geographically–yet each lot is given one ID and usually counted as one item in the collection. Wet specimens are sometimes in stainless steel tanks, but more often glass and plastic jars which are filled with ethanol or formalin. Cores and other geological specimens are kept in tubes, both PVC and cardboard. Storage of the containers the specimens are held within most often were open shelves, although some mention sets of drawers and closed systems. Fire code and specific safety issues came up with collections with wet specimens as the materials are very flammable. Most physical collections are intended to be kept in humidity and temperature-controlled facilities, though participants indicated this was not always the case. Climate control was most crucial to geologic core focus groups due to the nature of the specimens, especially the ice.

The duality of managing physical items and their digital surrogates meant all participants shared *digital storage* considerations related to the associated digital objects. These digital collections included a variety of files including a lot of photos and specimen imaging like CT scans, 3D modeling, datasets (e.g., bone measurements, core sample analysis), sound recordings, geographic data as well as field notes and pictures of the older specimen labels. The anthropologists focus group admitted most digital objects are unorganized. Organization tools mentioned included PostgreSQL-backed databases in the two geologic cores focus

groups, and most biocollections indicated using Specify, but Symbiota and SERNEC are also mentioned. Discovery is related to storage, but those indices will be discussed in a later section. These databases are both backed up off site (e.g., cloud) and on site in different combinations depending on the institution. Of note, the ice core computer is set up in the freezer for directly entering data.

Codes emerged relate for *file* or *data types and formats*. Data types discussed include visual files like jpegs, CT scans, TIFFs. Data was in Excel files, csvs, PDFs, and other flat files. Also mentioned were inaccessible file types. A few focus groups mentioned sound files (e.g., calling frogs) which spanned DAT format to mp4. The most unique file type was a .sdl mentioned by the managers of paleontology data that relates to 3D modeling.

## Data collection

Several codes emerged directly related to the data collection–*collection age* (17); *collection discovery* (48); *imaging-modeling tools* (8); and *software* (51). The *collection age* discussed in this study ranged in age from the mid 1800s for a herpetology collection up to 1974 for one of the geologic cores as the youngest mentioned. *Collection discovery* starts for a few reasons. Sometimes an institution of higher learning needs a research and teaching collection or a federal agency mandates a study. Collection managers frequently stated that the collections they now managed were started by a single individual, often a professor at the institution, or the institution itself and more specimens were added to each collection by many individuals and groups over time. Often these older objects were collected in eras without recent standards and/or laws to inform their systematic and ethical collection, especially taking specimens and items from countries and cultures. Most often the current collection is the repository for researchers at the institution to continue adding objects to. There were instances in focus groups that indicated some items come from outside the universities and agencies, such as the "Amateur Archaeology Society", retiring professors, the petroleum industry in geologic cores, exchange programs with other herbaria (i.e., loaning samples), NSF expeditions, state wildlife agency efforts, citizen scientists, and even things resulting from undergraduate classwork. Three of the ten focus groups mentioned absorbing orphaned collections from smaller museums or individual researchers justified by their potential value.

Several participants mentioned *image-modeling tools* to generate additional data from the physical collections. For instance, across focus groups some managers are using all types of imaging tools to learn more such as x-ray fluorescence, CT scans, images of histology slides, and even those in geologic cores procuring a hyperspectral scanner. Other *software* used to assist with managing digital derivatives included various local database tools such as File-Maker, Excel, PostgreSQL, TaxonWorks, "S3 Amazon compatible container," MinIO, Solr, Emu, Arctos, and Specify. Others included specific online databases or data repositories mentioned were Symbiota, VertNet, Encyclopedia of Life, iDIGBIO, FishNet, GBIF with Morphosource for images/imagining/models and GenBank for genetic sequences. All this organization facilitates discovery and relies on metadata and a few metadata creation software were mentioned.

## Metadata

As expected, metadata describing these physical collections and other digital representations are a critical part of curation activities. The metadata codes and counts were *metadata standards* (26), *locality data* (50), *GPS* (9); and *sufficient metadata for reuse* (33). Metadata are added to either a local database or repository software. The most common software

mentioned by several participants from herbarium, paleontology, and ichthyology collections was Specify. The relational database software, FileMaker Pro, was prominently mentioned by in the archeology, geologic cores, and ihthyology collections groups. The only specific *metadata standard* mentioned was DarwinCore. Most collections seem to have their own metadata standard for their collections, usually centered on the method of assigning ID numbers for specimens. A few systems were named (Cornell system, Nelson numbers, American Petroleum Institute (API) number) but not all are necessarily widely-adopted standards beyond their communities of practice.

Another common type of metadata stored is *locality data* that reveals the physical place and time the specimen was found, collected, and accessioned. Most collections stated that locality information is currently captured largely by GPS, but that items accessioned before GPS was in wide use tend to have spotty or unhelpful locality descriptions. Historic or legacy data in general had approximation issues across disciplines by either totally lacking a place name or any coordinates for a specimen. Archaeology and anthropology focus groups discussed locality data from the past as only having relative information like a general dig site or at most an area that was dug in a specific year. Some parts of collections only have descriptive locations (e.g., farm or county name). A common theme was having to hide locality data in some manner either for legal or ethical reasons as a way to protect specimens. All incoming physical items have a *GPS*. Ichthyology collections managers cited georeference tools such as GEOLocate, which is specifically used for natural history data. The research domain's history of extensive field notes, when transcribed and entered into databases also help with locality precision. In some instances, locality data is not publicly findable due to the necessity of protecting the locations where certain protected or endangered specimens were found.

Near the end of all focus groups collection managers were asked if what users are given is *sufficient metadata for reuse*. Like in prior studies, this DCP question is difficult to answer and responses ranged from a definite yes to plenty of nuanced conditions, and even a few no's. For example, the paleontology group stated for the vast majority of our users: no. Many reusers of physical collections want to examine the actual specimen for their research. A common perceived issue across focus groups was that the quality of metadata would improve across time as relatively more details are captured today compared to when some of these collections began. One exception was from entomology as they do accept donations that lack scientific metadata, but the collections still have value despite the absence of metadata. As these collections grow and acquire more items, the process of accessioning includes several steps as physical objects are ingested into them in an organized manner.

## Organization

From the accession phase, physical samples get assigned unique identifiers and organization begins. The organization section reviews coding related to *accession* (28), *collection organization* (56), which includes both the physical and digital organization of the collections, *persistent identifier* (PID) (42), and *backlog* (12). Each type of physical sample presents its own unique challenges to organization and approaches across disciplines varies. Most collection managers mention a numbering system of physical objects that then is mirrored when possible in the digital knowledge representation.

The first phase of the organization of these collections in the acquisition process is *accession*. The term varies across focus groups, with only three of the ten groups using the exact term, but any discussion related to taking something into a collection was given the code–*accession*. In this phase of an objects' life cycle, items in the collections are inventoried and may be assigned an accession number. In many cases the number may include the assignment

of a *persistent identifier (PID)*. Some participants reported that PIDs given did not follow a universally accepted standard, instead adhering to an institutional convention (e.g., FMNH PR 2081). As mentioned, this number typically included the accession number and may have also included identifiers for locality, storage location, or when applicable taxon.

Some focus groups mentioned that the numbers had meaning that related to the year acquired (Archaeology) or family (Ichthyology). The ichthyology focus group mentioned the Nelson number, a herberia focus group named the Cornell system for fungi, and geologic cores uses the API number common for oil and gas. The geological collections use a variety of organizational techniques, but all start with standardized numbering for metadata (e.g., API). For biocollections, the most common approach is the "Darwin Core triplet. So, an institution code, a collection code and a catalog number" (FG5-A).

The archaeology focus group participants mentioned broad organizing tools used depending on the items needing to be organized (zoological, archaeological, geological). Archaeology collections provide a microcosm of all physical collections because they may retain many types of physical samples related to the same event or instance. Other physical species collected do not retain their surrounding ecosystems, but simply the single specimen. Within zooarchaeology, a taxonomic system and other archaeologists mentioned numbering systems that mapped to specific years or dig sites. FG1-C explained the workflow of aggregating data across archaeology as "knitting together a bunch of different standards for different components of what you're trying to record that ultimately end up in just like a gigantic, gigantic data recording form." An information scientist might call this a metadata crosswalk, but a lack of standardized terminology for managing physical and digital collections across domains complicates data interoperablity.

FG9-A summarized the accession process for a frozen carcass "we preserve it, give it a curation accession number for the collection, verify the data that the whoever is collected it has given us, put that onto a physical worksheet as a paper backup, and that becomes that physical worksheet is then digitized when we push that information into Arctos, which is our collection management software that is done by curation assistants typically graduate students." The duality of tangible and digital curation may double the workflow, but is essential to maintain provenance and facilitate reuse, AI readiness, and public access. The records management schedule of uploading data to domain repositories and metadata to indices varies, but this work to share with larger indecis is done. "It's usually only once a year. I try and send a data upload to the big data aggregators" [FG10-A]. So, as items are accessioned locally and made more widely discoverable by sharing the metadata as FG3-A stated "it's both on our online database and then gets mirrored in iDigBio and Darwin Core."

For biocollections most, *collection organization* is informed by Linnaean taxonomy. Generally starting with larger taxa (frequently family but in the entomology focus group they mentioned orders) and going down to genus and then species, many specimens would also be organized by location of collection. Some herbaria collections first use geography to sort and then by others in the same group do it the opposite way (i.e., species then geography). FG7-A explained their collection was "alphabetized by genus, and then species, and then geography." Another herbaria manager explained that they "organize our vascular plant collections alphabetically by family according to the APG Group, the Angiosperm Phylogeny working group" (FG6-A). One paleontology collection manager, FG3-A, explained "it's broken down into our 3 major collections, which are paleobotany, fossil invertebrates, and fossil vertebrates, and then everything's organized stratigraphically within those groups, so individual cabinets are locality based rather than taxonomy based." The intersection of formerly living organisms and rocks presents an interesting slice of how each physical sample's unique makeup determines

the facets most useful to organize the collections. The different approaches to *collection organization* even within the same collection types result from local solutions to many inherited problems; however, in digital spaces if the original order of physical collections is irrelevant as digital objects may be manipulated and sorted by any facet.

Geologic cores, although all similar objects, arrive at the collection through a few ways, including collected by the US Geological Survey (USGS), a university partner, an NSF-funded project, or privately drilled and donated. They are organized by geography or accession time, but how they come to the collection organization. An unavoidable issue across biocollections are the donations from citizen scientists, which run the gamut of quality, but even if inconsistently acquired do bring value to existing collections. For these objects, a lack of organization upon accession impacts how they may or may not be integrated into existing collections organizational structures.

The DCP questionnaire included a direct question about collections using persistent identifiers and that code emerged in all groups–*persistent identifier (PID)*. In geology, the International Generic Sample Number (IGSN) is the widely used PID, but more work is needed to go back and create identifiers for older items. As a geologic core participant put it, the subsequent organization stems from this original decision "the core that was drilled in the Western Antarctic Ice Sheet Divide was called the WDC for Western Divide Core, and then O6A because it started drilling in 2006, and it was the primary core, so it got the A designation. There's WDC O6A, B, C, and D" (FG4-A). The unique IDs signify other meanings and knowledge organization approaches vary. For example, a few simply assign a unique ID in the order that something was added to the collection. One participant mentioned the name begins with the acronym for the university (FG9-A). All herbaria have a unique letter code in the Index Herbariorum that is globally recognized (e.g., K for Kew Royal Botanic Gardens). Then each item gets a "L for lichen, or B for bryophyte, or M for mycology" collection (FG6-A). This allows the circulation of materials without losing their original description and reduces the risk of duplication of any numbers or items with multiple persistent identifiers.

Some assign DOI's to the collection, project, or item level. This largely depends on data publishing requirements across fields, which may mandate objects have a persistent identifier to be cited. The individual practices of each manager and their availability also dictate how detailed collections may be. FG1-B from archaeology, indicated that "every specimen has a DOI, they can be downloaded; they can also be cited individually so that you can cite a single bone by itself." Item-level persistent identification was rarer in biocollections focus groups where aggregation happens at various levels (i.e., grant, project, so forth) and many things may be lumped together and given a single DOI. In geology, APIs were mentioned for cores, but several participants also listed IGSNs. For cuttings, each new sample is unique and given an internal identifier separate from the original sample, but not a formal IGSN. This approach differs from biocollections as often when there is more of the same thing (e.g., tadpoles) the more likely the species are not to have a collection level ID. Also, individual specimens that have related DNA, tissue, and so forth all connect back to the original specimen ID. For example, a single DOI may represent "15,000" tadpoles (FG10-C). This approach allows for multiple access points to know where is what and what is where. One participant summarized the cyberinfrastructure succinctly "*Specify* produces GUIDs for all of the objects in the collection. Not only collection objects, but also localities, agents, taxa, you know, all of the major tables in the database have GUIDs associated with them that are all published out as part of Darwin Core to all of the aggregators. Our database–our database as a whole, then, also has a GUID associated with it when it gets published to–to GBIF" (FG5-A).

Although most of the references thus far are to persistent identifiers, focus groups also mention the use of curation numbers or tags to serve as a link to physical objects. Like the

digital surrogates of these objects, the storage space itself informs some decisions in organization. FG4-C stated "we're able to tilt on their side and slide into the shelving much like, um, library books, actually." This is the case for ichthyology collections where phylogenetic order by Nelson numbers and alphabetically by families was common, but a few collections running out of space have to be creative. One solution is sorting collections by jar size to reduce wasted height between shelves. Automated storage systems exist in libraries and archives. Specialized robots retrieve boxes where documents and books are located and track where each item is taken from and where things are put back. Therefore, the order losing meaning since physical collections are shuffled around based on frequency of use. What is lost through serendipitous discovery of similar samples located near each other should be accounted for if similar changes occur in all scientific physical collections as the efficient use of space justifies this rearranging, but may have unintended consequences removing accidental discoveries.

Retrieval in collections uses similar approaches to libraries as well, with each row and shelf retaining an address to look up where an object should be located. One issue with any physical organization system is change. For example a herberia and the ichthyology focus group participants mentioned being locked into previous taxonomic divisions that were no longer accurate due to phylogenetic changes. As FG5-D put it "we've got a legacy system where everything was numbered by families in 1966. And we have to stick with that [. . .] So, that's how we're stuck." Still, there is hope for some items not organized getting reused with new tools to make their data more findable. FG10-B explained that histology slides have been "uncataloged and sort of uncataloged and housed in slide boxes" but now due to the use of MorphoSource these hard to locate items now may be much more easily cataloged and located with these database tools.

Most participants are behind on knowing what all is in their physical collections, let alone having a detailed record of all their digital surrogates. Every focus group had most participants acknowledging some sort of backlog. *Backlog* became a code assigned to the items not yet processed into collections. An extreme example from the entomology focus group "after 30 years of digitizing, we are about a third of the way through digitizing our collection" (FG9-B) gives some idea of how backed up the backlogs might be. For insects, catching up might not be possible as FG9-B states "I don't have time, and I also don't even have the expertise to verify every single specimen as it comes in because there are so many different kinds of insects that it would be. It's pretty much impossible for me to verify every specimen." In herbaria, there are similar issues and all collection managers of these types of physical collections are behind in different ways. "We've got a backlog of maybe 4 or 5,000 that we're working on" and "less than a quarter of the collection is photographed" (FG6-A). FG6-C "we also have a backlog of probably 10,000 records." Backlogs across domains face different challenges as some samples deteriorate faster than others. FG6-B explained there are "unprepared collections in newspaper in like big Tupperware bins that have some silica gel, and that have been duct taped to try to preserve the specimens until we can get them prepared." Many reusers of data resulting from these physical collections may not realize the great efforts taken in advance of anything becoming findable.

## Findability

Codes related to findability include *collection manager* (19), *accessibility* (7), *data portals* (21), and *findability* (41), a catchall for general comments related to the broad concept of making things findable. Although collection managers are individuals, in many instances they are *the* resource that knows what might be where in their collection. Collection managers' institutional memories could be recorded in some manner, perhaps using AI tools to gather oral

histories, but if not done may be lost in each departure of staff. Limited staff partly explains the backlogs of most physical collections. Staffing volume ranged from a single full-time staff member, a part-time staff member or one individual managing a collection, but is also a professor with classes to teach! More resourced institutions may have teams ranging from a couple to at least five full-time employees, with all places explicitly mentioning a mix of students, graduate and undergraduate, as well as part-time staff, and volunteers, all contribute to keep things available. This smaller staff sizing could lead to individualized and institutionally specific approaches despite organizations and standards available across domains. As FG3-B put it "I am a one-woman show. It is me, and I, I am the controller of the collection" and ultimately the individual managing accessibility to these collections.

The code *accessibility* captures participant sentiments related to making their collections, both physical and digital, accessible. Although format obsolescence is a known concern there may not be solutions for every manager, as FG2-B put it "we have data in every type of format starting from the 1970s up until present, and not all of it is accessible to us today, so that is also an issue we're trying to deal with." Accessing older versions of data and software present data curation issues for physical collections. The stories of migrating older databases to Excel files then to a "real" database present a necessary hurdle to preservation with each data migration propagating risk of loss through data transformation and integration. Maps, drawings, floppy disks, film, punch cards, all require managers to keep older computers just to access those otherwise unviewable things. *Accessibility* also includes all the positive mentions from focus group participants making physical collections ultimately more reusable. The dream that anyone in the world with Internet access may download a ton of this data and digital objects motivates all of these collection managers. An issue raised by one participant is that users pulling data from their local database will have the most updated version, but any users pulling specimen data from other aggregators might be downloading an older version as some portals do not pull updates often. This issue may cause issues for reproducibility in science given the data pulled could be different each time. Luckily, data aggregators do contact collection managers and vice versa when issues are discovered, mostly by data users. As FG10-A stated "the ones that are managing these things like they are overworked. Like it, it takes a lot of time to do this stuff, and they have to do a bunch of data validation." There are a lot of steps and a lot of potentiality to make mistakes, but all these portals are still reliant on human power and very little is automatic whatsoever.

*Data portals* emerged as its own code given many were named by participants. Biodiversity management products mentioned included Symbiota (8), Global Biodiversity Information Facility (GBIF) (6), Specify (5), iDigBio (3), VertNet (3), Arctos (2), Bryophte Portal, MyCoPortal, TaxonWorks, FishNet, and MorphoSource. These tools are essential as that is how collection managers communicate with users and each other. Many of the portals listed overlap and this increases accessibility to these objects. Most participants mentioned having their own searchable databases hosted on websites, but the aggregators are key to reuse. FG10-A reminds the group "we actually have a physical server on campus that's backed up, in a few places every morning" to underscore the relation of storage with access. Data portals increase access to biocollections and the number of them may need some explanation. As one ichthyology participant put it, "our portal has fields that, for example, GBIF doesn't show under, on their results" and the different views assist the user experiences [FG10-B]. Four of the ten focus groups were not biocollection managers and it is unclear why specific portals were not named in those groups. Perhaps, forensic anthropology, archaeology, and the two geologic cores groups do not have the same demands for their collections and related data to be accessible via a portal.

All other mentions without a more specific concept were coded as *findability*. All focus groups were asked "Is your collection findable?". FG2-A had a refrain shared by the managers of more protected collections, "No. No, not at all." Several of these physical collections, at least large portions of them, are only findable through physical access to them. Collection level metadata allows potential users to know general information that a collection of objects exists, but not at a granularity to explore particular items, species, and objects. Some collections have additional barriers to access. For example, collections that have human remains hold agreements with donors that limit specific information that can be shared. Public access does not exist, and potential users must reach out to specific collection managers to learn about their holdings. In archaeology, an interesting distinction was shared "what's discoverable to the general public is our looted and fake objects" (FG1-A). This accessibility continuum from human donors with rights, to human artifacts with cultural value, to species with additional preservation care, to non-living items, follows somewhat from very closed access to increasing availability. Even a casual museum goer can touch a dinosaur footprint, but many other materials are behind glass, handled with gloves, or not viewable at all.

Many other physical collections are somewhat findable due to the expressed goal to "make sure that all the collections are accessible online" (FG1-B). Accessible but not necessarily findable was a theme throughout the groups. Several participants expressed that items could be accessed, but admit they are not as findable since potential users would need to know who to contact or where the items may be. These barriers to AI readiness exist where a human may navigate personal networks or recall locations of certain repository names from past searches, a machine would need additional context through metadata to access items.

In archaeology, several mentioned Open Context as a way others could find items; however, many more objects which are not indexed are not very findable without human intervention, assistance, and effort to locate these discoveries. Data entry errors were mentioned as a potential concern as rushed items made findable may actually have inaccurate metadata and users find things that are mislabeled such as "a large stone object was listed as antler for years" (FG1-D). Participants explained that the control of the collection and who has permissions to "see it, study it, and publish it" (FG1-A) determine how findable or discoverable any physical object is. Even when there are no limits to who can access something, findability still requires some metadata and organization to be found. As FG8-B explained "we were given a lot of things in the past that didn't have metadata. And we have to open each box and find out what's in it." For items that are cataloged, users can just search and find those objects. One geologic core manager explained there are several different search parameters that can be used, facets such as geographic locations, geologic formation, stratigraphic information, depth, state, API number, collector/operator name, and other keywords. Geological portals have a map-based web interface to facilitate searching by location. For federal agency collections, items are findable through multiple aggregators such as the US Geological Survey's Science Catalog. Findability is increased by pushing databases to other shared databases especially across biocollections as each of those focus groups mentioned domain aggregators (e.g., Arctos, Specify, Symbiota, iDigBio, GBIF, MyCoPortal, SERNEC, VertNet, FishNET, and others).

Many mentioned findability in relation to institution's websites. Assessing the information retrieval and usability issues across all of these platforms presents another research avenue to increase and improve findability of these physical collections. Participants explained that data is shared in multiple places because each allows users to search different metadata fields and user needs vary that may determine which portal they prefer to find data through. FG7-A stated the obvious "not all of them are database yet" and with backlogs and limited resources chunks of these herbaria require any user to "literally have to go to the cabinet and open it up

and look and see if there thing is there." FG7-A elaborates "you have to talk to a human who explains to you how to find things." Clearly, the state of those items is far from the machine-actionable data called for in the FAIR Data Principles. The goal to have every object findable was echoed in all focus groups and as herpetology focus groups participant FG10-A put it "I'm working on it." Of the various physical collections, biodiversity drives a diversity of potential uses that may have led to the variety of ways to find these particular items.

## Reuse/Use

The distinction between reuse and use can be difficult to parse out for any collection manager, potential users, and even these authors. All the management efforts summarized thus far in the responses to the DCP questionnaire lead to enabling reuse of these physical collections. The codes for this section do not differentiate between use and reuse as these terms were used interchangeably in participants' discussion. The reuse/use codes and counts are as follows *users* (41), *uses* (46), *teaching uses* (16), *outreach* (12), *use constraints* (50), *destructive sampling* (19), *citation-credit-acknowledgment* (15), *repatriation* (2), *license* (3), and *value* (5).

The types and amounts of potential users (41) named from forty-three collection managers across the disciplines of anthropology, archaeology, botany, geology, ichthyology, entomology, herpetology, and paleontology are vast. Nearly all academic institutions list faculty, staff, all types of students, postdocs and other visiting researchers. Particular types of scientists were also named–ecologists, paleontologists, zooarchaeologists, state and regional botanists, paleoethnobotanists, and paleo climate researchers. One reflection from a participant in geologic cores assumes their users are "primarily all academics that use this facility because they know that it exists" (FG8-A). This is a valid claim as many of these collections do not widely market themselves and awareness may be limited to word-of-mouth within communities of practice. A few indicated they serve users worldwide as their collections are globally unique, but most acknowledge the order of magnitude of users shrinks as you move from those with regional, national, to global scopes of their research. At least one participant mentioned that the pandemic lockdowns and travel restrictions increased international requests about their collections and resulted in sending more data to foreign researchers.

Specialists and teachers in many of these areas beyond academia were listed as potential users. Many mentioned k-12 teachers and their students. A few biocollection managers mentioned some new kinds of users–the pre-student and hobbyists that seek certain types of specimens in their collections because they no longer can be found in the wild. This type of user underscores the importance of a better understanding the lifecycle of these collections as their curation keeps a record of life and the abiotic world as unspoken cultural heritage objects for any resident of Earth beyond all the scientific users. Anybody interested in the study of these domains can be a difficult community to scope if any individual or machine could be considered a reuser of these collections. The term *general public* was mentioned three times. A herpetology manager provided a great spread of the variety of general public "we do get private researchers who are like guys who work on taxes by day and turtles by night. It's so folks of that sort, retirees of a similar boat" (FG10-C). Finally, some media were mentioned that may use museums in documentaries and news stories about research in these areas. One mentioned their local Art Museum uses their collections if any relate to exhibits or artists need objects for models to paint. Of note, none mentioned machines or AI as potential reusers of these collections, but clearly in today's more open, FAIR-aligned, data-driven sciences machines are reusers.

Quantifying users is key to many organizations to tailor services and resources to meet their needs. In other traditional information agencies, knowing counts of users is required to

justify expenses. In future studies, collection managers could take time to provide more precise estimates for their users but in these groups there were only estimates. For example from the geologic cores, "I'd say 80 percent of our users are from the petroleum industry. Probably 15% are students or professors from academia. And the other, I guess minerals—critical minerals. Professionals" (FG8-C). Another geologic core manager provided a similar distribution with the addition of government researchers (e.g., USGS scientists, state wildlife agencies, and so forth). The geologic core managers are unique in that they curate items, but analyses of their collections occurs elsewhere. Perhaps, that leads to better tracking of the uses of their collections where the analyses take place.

A plethora of *uses* (46) were mentioned. Foremost is knowing who has what where. Many items in collections have multiple uses as demonstrated in this comment by an entomology collection manager "biomimicry research by examining different morphological aspects to engineering research by looking at different properties of different insects" [FG9-B]. Research is the primary reuse discussed, but most also mention educational purposes for their collections with particular collections dedicated to teaching uses. Reuses may help reinterpret old results or to fill in knowledge gaps with new information. Research uses span the gamut of scientific inquiry from common metrics of all types, to other types of analyses, including genomic (e.g., ancient/modern DNA), 3D and CT scans, x-rays, dating a site, understanding glacial histories, isotope analyses, analyzing the chemistry of items, carbon sequestration, energy assessments, mineral exploration, analyzing carbon and nitrogen phosphorus sort of ratios, morphological measurements, phylogenetic work, exploring pharmaceutical users, inform conservation planning, species count, specimen distribution, species name changes, managing invasive species, protecting commercially viable species, other ecological studies, modeling of all sorts, and even looking at "gut contents." Further osteometric data (i.e., bone anatomy) provides context for identifying species and may give insights into the climate they may have lived in. A few other uses mentioned included creating a state atlas of amphibians and reptiles, art projects, and comparing their holdings to other collections. A paleontology collection manager mentioned three users that need forms for destructive sampling, reproduction, and imaging, but certainly if probed others may have processes and procedures that are more formal to facilitate use.

*Teaching uses* (16) were mentioned by most focus groups. Instructors use their collections in coursework or labs associated with classes at institutions, high school summer classes or camps, as well as other training for specialized users or educational outreach events. Since the groups covered many domains, the courses mentioned match that diversity–forensic anthropology, paleopathology, biology, taxonomy, botany, ichthyology, paleontology, evolution classes, and even one literature class looking into ancient herbals. Some mentioned separate teaching collections for these purposes. With greater access to these collections through digital surrogates, the educational impact is also bigger with educators all over the world.

The theme of *outreach* (12) closely aligned with teaching uses. As mentioned, the teaching uses of collections often include events beyond for-credit instruction. Art was mentioned a few times in herbaria and ichthyology groups, which included growing herbarium specimens then carving them into metal as art, and another event had the managers hang real specimens next to Audubon prints. The herbaria participants also mentioned tours for kindergarteners, middle school STEM students, local community college students, and even master naturalists. Whereas herbaria participants all indicated visitors coming to the collections, the herpetology and paleontology groups described taking items to schools, science fairs, street fairs, and public libraries to engage communities in appreciation of their region's natural history heritage.

The most coded for all topics within the *reuse/use* section is the broad concept of *use constraints* (50). The importance of constraints on usage being discussed more than actual uses (46) throughout these focus groups highlights those protections on the access and use of these valuable objects remains as vital a consideration for managers as simply using them. Not surprisingly, the focus group of forensic anthropologists mentioned restrictions on destructive analyses that include an approval process where a committee decides what is allowable. Overall, due to these being human remains, photography is not allowed. In archaeology, human artifacts retain levels of approval for any use of resulting data, including permission from "the excavators, but also from the museum as well" (FG1-A). Locating these artifacts is difficult and timely, therefore it is not surprising that those making the discoveries have been very protective of their capta and data to ensure they are the first to publish from it. Complicating matters of use further is in archaeological collections intellectual property rights differ across nations. For example, "In England, museums have copyright on their materials, and when you publish them, you have to copyright them" (FG1-A). In other countries, "local laws precluding transportation of, or exportation of, animal bones or other archaeological objects from leaving the country" differentiating ownership from possession (FG1-B).

A similar pattern of first to discovery, first to publish, appeared in the geological core focus groups. The National Science Foundation's data use policy indicates that "collectors get first dibs at getting data off of the samples for 2 years, and then everything becomes public" (FG8-A). If you drill the core, then you have exclusive rights to analyze the core first and this was similar for other geologic samples discussed (FG4-A). Different types of data and analyses have their own unique license agreements "2D and 3D seismic" include any reuser to share digital objects back with the loaning institution (FG8-B). Timing and volume for geologic cores has a lot of restraints mentioned by the groups, including not oversampling a spot and returning any data from analyses of a sample within six months (FG4-B). For anyone violating the policy, those users are not allowed to return to use anything again.

In entomology, restrictions of use include no destructive sampling, and users are not allowed to "destroy them in any way, or degrade the integrity" in whole, but some DNA sampling may include the destruction of part of the specimen (e.g., hindleg of a grasshopper) (FG9-A). Similar to other physical collections, loan agreements for entomology items request acknowledgment in publications to track and report the use and in some instances request authorship. This includes citing DOIs that were created in GBIF. Unique to entomology collections is that these items do not circulate, but must be analyzed on site. For the herbaria focus groups, some mentioned not loaning to individuals but only to other institutions with "controlled facilities and has sort of an established reputation" (FG7-A). The same participant also explained that "specimens on the rare plant list for the state or federally listed as rare, are obscured" and the data is not released of the exact coordinates where they were located. Commercial interest foragers that are only interested in the locations of valuable species such as ginseng are turned away, but since the herbaria are "partially publicly funded institutions" the public are welcome by appointment only (FG6-A). Several mentioned education and research as the foci of these herbaria, so actual specimens are not shared with bioengineering firms. Still, images of specimens are openly available.

For herpetology collections, similar restrictions were mentioned related to protecting endangered species and not allowing anything destructive like 'examine its gut contents" (FG10-A). Participants acknowledge that enforcing the attribution policy is a challenge but expected any user to cite the loaning institution. Like the concerns over obfuscating localities for valuable plants, herpetology managers obscure locality data of endangered specimens. As FG10-A explained "some of them actually make a good bit of money poaching these animals and sending them across the US and around the world." A dichotomy exists across all

physical collections that one participant put succinctly as "the eternal battle between conserving and using the specimen" with a reframing that any destruction of a species that leads to discovery needs to be shared through any number of the data portals and the loaning institution (FG10-B). Additional protections are put in place for any extinct species. The wet collections of herpetology and ichthyology have similar considerations for sharing species that are in liquid that impact shipping. Use constraints mentioned by participants in ichthyology match others mentioned through the use of CCBY licensing, such as commercial restrictions, no destructive sampling, no giving to a third party, and attribution for any use of species in publications. Like all other physical collections forms exist to inform use with specifics on what and how to do certain analyses for example "if they're making a study on diet data, we makes sure that they know how to preserve the gut contents so that we can get them back" (FG5-D).

Use constraints in paleontology include many of the same aspects discussed prior, destructive sampling, reproduction, imaging, and commercial use (FG3-A). Commercial use is a particular concern given some fossils may be reproduced for display and have real value, but that must be balanced with the intent to make 3D models as accessible as possible to "print them for classes or outreach" (FG3-A). Protection of the tangible objects include only loaning fossils to advisors and not individual students to avoid loss (FG3-B). One loss built into the nature of some methods is the actual demolition of objects and deserves special discussion–*destructive sampling* (19). Whereas any geologic sample requires some type of destruction to gain new knowledge (i.e., breaking rocks apart) as one participant put it "most of our analysis is destructive just because of the nature of it" (FG4-A). At the other end of the continuum would be human remains that require additional justification for any destructive analysis. For forensic anthropology collections, a committee assists with the decision to allow destruction or not and all others the choice is made by the director or curator of a collection. For others, new knowledge requires using new tools and techniques and might be the only way to gather important information like ancient DNA. Entomology falls on a different spectrum as most said they do not allow anything that would "compromise their long term storage and use" and at most one could destroy part of a specimen [FG9-A]. The same limitations are used in herbaria as one participant explained "if there's enough material sufficient for destructive purposes" but often there is not enough left to do so (FG7-B). In instances where destruction is done, all items are additionally documented with more pictures and most participants mentioned having use agreements to send back leftovers. In the case of DNA from destructive sampling, findings are shared with GenBank. An additional step for all involved, but essential to retain the invaluable materials as any removed material belongs back in the collections for future preservation and discovery. Collection managers are fairly risk averse knowing the value of their items as one remarked we "really want a viable reason why you want to chop them open, remove gonads, blah blah" (FG10-C). Most participants mentioned restrictions on third party destructive sampling to reduce the risk of loss without some new data gained. These unique aspects to physical collections differentiate the curation of tangible objects compared to most digital counterparts and the process and findings of destruction should be a consideration informing the design in capturing provenance in cyberinfrastructure.

Appearing in the use restrictions enough to be its own code, *citation-credit-acknowledgment* (15), appeared across anthropology, geologic cores, entomology, herbaria, herpetology, and paleontology focus groups. Since budget justification of some collections relies on metrics such as citations, several participants shared concern that although "researchers who come are supposed to share whatever data they collect" many speculate that it does not occur and cannot be enforced (FG2-C). Unique to human remains is that specific donors cannot be used in

publications and must be further anonymized. The majority of physical collections do expect users to both acknowledge the repository an item comes from and send copies of any data resulting from analyses afterwards. Since many participants pointed out managing this workflow as a limitation, future cyberinfrastructure may build in better tracking of these local processes to ensure attribution does occur. As one geologic cores participant put it "it's kind of like the honor system. Yeah, unfortunately it's not, you know, regulated or enforced" (FG8-C). Perhaps, publishers and funding agencies could assist compliance by tying publication or future funding to demonstrated compliance of attribution. As FG8-A states that each and "every single sample request, has some language for them to put in their acknowledgments about use of the facility" to make it easier for researchers to acknowledge collections but also expressed "I can tell you the compliance is minimal." In entomology, loan agreements include not only the acknowledgment language but also ask researchers to "please forward us a copy of your paper so we can add it to our library" as another means of measuring impact of collections use (FG9-A). In paleontology, loan forms also include "what you need to credit, and how you need to credit" institutions (FG3-B). As mentioned, prior, the entomology focus group highlighted that they "found to track that better using GBIF's built-in citation trackers" that gives all a doi to cite (FG9-B). Herbaria participants were most optimistic about acknowledgments even explaining how it is part of the culture "it would be very poor manners if you did not acknowledge" and that community of practice has attribution engrained (FG7-A). Loan letters in herbaria also outline expectations for "being sure to deposit your things in GenBank and connect them together" (FG7-A). Each type of physical collection has different levels of compliance but all the same sentiment that citation-credit-acknowledgment is expected because that "helps to justify to administrators that our collections are valuable" (FG6-A).

The last set of codes related to use/reuse appeared only a few times but were all unique enough to call out and describe. *Repatriation* (2) is a consideration for anthropologists and paleontology collection managers. "We are in the process of repatriating a large number of them due to Federal law" was a refrain from a participant on returning indigenous peoples to their land as opposed to the collection their remains are currently kept in (FG2-B). Although human remains are guided by national and international legal considerations, many collections now reflect on the potential return of items to their places of origin. Efforts discussed in the paleontology group include "tying our collections back to the communities from where they came from" (FG3-C). The example provided was giving back a dataset or 3D model and not returning literal items, but at the very least expanding the interpretations of items to include indigenous perspectives.

Another intellectual property facet mentioned a few times was *license* (3). Throughout use constraints discussions various licensing were mentioned, but in herbaria the data portals mentioned of GBIF and SERNEC indicate that a creative commons license is chosen during data deposit. The most mentioned license throughout all groups was the CC0 waiver as it applies to most images of physical collections and if there are "other restrictions on that particular item, or specimen, or collection type" those are clearly stated (FG3-C).

Finally, *value* (5) appeared to be a consideration mentioned in the entomology, herbaria, and paleontology focus groups. With extinctions on the rise, one collection manager noted that "seeing that it is not so easy to find certain things, species in the wild anymore… I think this will make our collection more valuable than before" (FG9-C). Conversely, others expressed concern of "the lack of knowledge of administrators understanding the importance of these collections, and how useful they are" (FG6-C). To echo the discussion of value

demonstrated by citations, attribution, and acknowledge, one paleontology collection manager stated "we are constantly trying to prove our purpose and existence on campus and getting people to research and publish on our specimens is the number one way to do that" (FG3-B). Despite the complicated variety and volume of physical natural history collections as well as the complexity of their organization, access, and use, these focus groups gathered the perspectives and behaviors from current managers across disciplines. A brief domain-agnostic discussion follows to inform the design and implementation for an AI ready cyberinfrastructure needed to advance understanding of all earth science data-biological and geological.

## Discussion

One assumption of this study and its participants is that all physical collections retain the inherent purpose to be reused. Across all focus groups the foremost user need mentioned is knowing who has what where. It may benefit future work and study of physical collections to unpack the purposes of metadata. In 1904, Charles Cutter outlined a few purposes for metadata of bibliographic records—–(1) "to enable a person to find a book of which either the author, title, or subject were known"; and (2) "to show what a library has by a given author, on a given subject, or in a given kind of literature" (p. 12)m [23]. In library science the 1998 (IFLA) Functional Requirements for Bibliographic Records (FRBR) specified metadata purposes to find, identify, select, and obtain items for all sorts of reuses [24]. Differences between books and the physical objects discussed in these focus groups (e.g., rocks, human artifacts, and remains of living things) are obvious, but the metadata facets needed to find, identify, select, and obtain any information are strikingly similar.

Regardless of the object, users may or may not know what they are looking for and metadata must address both the user that knows what they want and those that do not. The focus group participants listed all sorts of potential users and user needs from every resident of Earth curious about the world around them, to those teaching about the world, to those highly specialized researchers doing science across all domains and institutions. The real purpose of knowledge representation and information organization is to connect people with the information they seek in a structured manner. Physical collection managers face many obstacles to building a cyberinfrastructure atop the various sorting and backlogs across all types of physical collections and the myriad of items. Results from the focus groups revealed multiple issues in curation behaviors and perceptions of physical collections managers that hinder reuse and lower the potential value of these items. The most mentioned challenge is the lack of standardization across collections, but especially within fields regarding the metadata and repository software used. Standards and wonderful interoperable tools exist for data discovery and reuse, but adoption and updating older items and related data take more resources than provided. Without more staffing of a data-skilled workforce, with knowledge of metadata representation in addition to all the other abilities needed to maintain physical collections, the findability and reuse potential of all collections suffer. An example of remedying this comes from the herbaria collections where all participants uploaded information about their specimens to the same data portals resulting in rich multi-faceted uses of their collections. Each domain continues to strive towards a more organized set of collections and the metadata facets expected by both human and machine reusers.

The external forces from various data movements, publishing requirements, and data sharing and management plans demand continued improvements. This study should highlight that many physical collection managers will need additional funding to upskill current workers with data and AI skills as well as hiring staff with this expertise. The dual costs of keeping current collections and their digital counterparts both as FAIR as possible for AI readiness

and accessible to scientists that must use the original items increase the need to justify these expenses through more consistent attribution. Focus group participants call for a cyberinfrastructure that better tracks uses and helps ensure attribution occurs.

For both tangible items and their digital derivatives, focused attention on metadata increases the reuse potential and subsequent knowledge found through use of these item by both human and machine users. The facets such as time, geographic locations, geologic formation, stratigraphic information, taxonomy, collector name, and other elements used to describe all objects of the world is mind boggling. Still, similar facets exist across sciences to inform more consistent description of items. Of note, use constraints receive more mentions in this study than actual uses. The preoccupation with constraints on usage highlights the importance of these protections on the access and use of these valuable objects. Additional metadata and consistent use of data licensing for any derived data should assist with balancing the necessity of protection with the need to reuse. The accessibility continuum from human donors with rights, to human artifacts with cultural value, to species with additional preservation care, to non-living items, could be explored further.

The study has limitations like all focus group data collection as more willing participants for this type of study may also be more motivated to make their collections as accessible as possible and not representative of practices across all physical collections. Anthropology, archaeology, paleontology, ichthyology, herbaria, and geologic cores although a broad range of sciences still lacks a few other physical collections considering a variety of other types of geological, biological, and archaeological physical items exist. Future studies should explore other collection manager communities of practice and attempt to conduct more quantitative data on others' data curation perceptions and behaviors.

## Conclusion

This study investigated the perceptions and behaviors of physical collections managers through focus group discussions from anthropology, archaeology, paleontology, ichthyology, herbaria, entomology, herpetology, and geologic cores collections. The results indicate that physical collections vary greatly across fields and institutions in both size and age. The metadata collected for each item depends not only on the field, but oftentimes on the collection itself. These differences extend to the data repositories used to share collections and their level of accessibility. Increasing the standardization of metadata and data storage across collections of a similar field as well as reinforcing requirements for research to return results of their analysis of the collections' specimens should improve the accessibility and usability of these collections. This in turn could serve to increase the use and perceived value of these invaluable resources. Certainly, continued investment is also necessary to unlock these invaluable data using machine-actionable tools to advance and reveal knowledge still trapped within these physical collections. The Data Curation Profile method and other conceptual frameworks and even terminology from the interdisciplinary approach of information sciences could assist physical collection managers' understanding of the entire capta/data lifecycle and inform the design of a cyberinfrastructure to connect all the things.

## Acknowledgments

The study was partially funded by the Institute of Museum and Library Services' (IMLS) Laura Bush 21st Century Librarian grant program (RE-13-19-0027-19). The authors would also like to thank the various physical collections managers for their time sharing their expertise and experiences as well as Andrea Thomer for recruitment assistance.

## Author contributions

**Conceptualization:** Bradley Wade Bishop, Sarah Kansa.

**Data curation:** Bradley Wade Bishop, Jaxx Fox, Sidney Wanda Taylor Gavel, Emily Grace Chapin.

**Formal analysis:** Bradley Wade Bishop, Jaxx Fox, Sidney Wanda Taylor Gavel, Emily Grace Chapin.

**Funding acquisition:** Bradley Wade Bishop.

**Investigation:** Bradley Wade Bishop.

**Methodology:** Bradley Wade Bishop, Sarah Kansa.

**Project administration:** Bradley Wade Bishop.

**Resources:** Bradley Wade Bishop.

**Software:** Bradley Wade Bishop.

**Supervision:** Bradley Wade Bishop.

**Validation:** Bradley Wade Bishop.

**Writing – original draft:** Bradley Wade Bishop, Sidney Wanda Taylor Gavel.

**Writing – review & editing:** Bradley Wade Bishop, Jaxx Fox, Sarah Kansa.

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
