## [Decision Letter · Decision Letter 0]

24 Mar 2025

PONE-D-24-59078From Ice Cores to Dinosaurs: Physical Collections Managers’ Research Data Curation Perceptions and BehaviorsPLOS ONE

Dear Dr. Bishop,

Thank you for submitting your manuscript to PLOS ONE. After careful consideration, we feel that it has merit but does not fully meet PLOS ONE’s publication criteria as it currently stands. Therefore, we invite you to submit a revised version of the manuscript that addresses the points raised during the review process.

**Please address all the comments before re-submitting.**

We look forward to receiving your revised manuscript.

Kind regards,

Peter F. Biehl, PhD

Academic Editor

PLOS ONE

**Journal Requirements:**

1. When submitting your revision, we need you to address these additional requirements. Please ensure that your manuscript meets PLOS ONE's style requirements, including those for file naming. The PLOS ONE style templates can be found at https://journals.plos.org/plosone/s/file?id=wjVg/PLOSOne_formatting_sample_main_body.pdf and https://journals.plos.org/plosone/s/file?id=ba62/PLOSOne_formatting_sample_title_authors_affiliations.pdf 2. Thank you for stating the following financial disclosure: Institute of Museum and Library Services' (IMLS) Laura Bush 21st Century Librarian grant program (RE-13-19-0027-19). Collaborative Analysis Liaison Librarianship. (2019-2024).  Please state what role the funders took in the study.  If the funders had no role, please state: "The funders had no role in study design, data collection and analysis, decision to publish, or preparation of the manuscript." If this statement is not correct you must amend it as needed. Please include this amended Role of Funder statement in your cover letter; we will change the online submission form on your behalf. 3. Thank you for stating the following in the Acknowledgments Section of your manuscript: The study was partially funded by the Institute of Museum and Library Services’ 972 (IMLS) Laura Bush 21st Century Librarian grant program (RE-13-19-0027-19). The 973 authors would also like to thank the various physical collections managers for their time 974 sharing their expertise and experiences as well as Andrea Thomer for recruitment 975 assistance.We note that you have provided funding information that is not currently declared in your Funding Statement. However, funding information should not appear in the Acknowledgments section or other areas of your manuscript. We will only publish funding information present in the Funding Statement section of the online submission form. Please remove any funding-related text from the manuscript and let us know how you would like to update your Funding Statement. Currently, your Funding Statement reads as follows: Institute of Museum and Library Services' (IMLS) Laura Bush 21st Century Librarian grant program (RE-13-19-0027-19). Collaborative Analysis Liaison Librarianship. (2019-2024) Please include your amended statements within your cover letter; we will change the online submission form on your behalf. 4. Please include your full ethics statement in the ‘Methods’ section of your manuscript file. In your statement, please include the full name of the IRB or ethics committee who approved or waived your study, as well as whether or not you obtained informed written or verbal consent. If consent was waived for your study, please include this information in your statement as well.

**Additional Editor Comments:**

Please address all the comments before re-submitting.

Reviewers' comments:

Reviewer's Responses to Questions

**Comments to the Author**

1. Is the manuscript technically sound, and do the data support the conclusions?

Reviewer #1: Yes

Reviewer #2: Yes

2. Has the statistical analysis been performed appropriately and rigorously? 

Reviewer #1: Yes

Reviewer #2: N/A

3. Have the authors made all data underlying the findings in their manuscript fully available?

Reviewer #1: Yes

Reviewer #2: Yes

4. Is the manuscript presented in an intelligible fashion and written in standard English?

Reviewer #1: Yes

Reviewer #2: Yes

5. Review Comments to the Author

**Reviewer #1: **This is a very useful study which will contribute to our understanding of the problems facing collections' use and reuse. I recommend that this article is published basically as is, though it does need a careful check for the few remaining typos.

May I recommend to the authors the following for future research and dissemination of these findings? First, I would recommend that the authors look further into the issues raised at the end of the Metadata section (Lines 333-344). The issues of sufficient metadata, usability of metadata for diverse research queries and standardisation remain vastly under-researched. The fact that this was a major concern of the Collections Managers shows that this also remains an under-researched issue.

Second, may I suggest to the authors that they consider presenting these results, and attending, the following joint conference of The IEEE International Conference on Cyber Humanities (IEEE CH), to be held in September in Florence, Italy. The subject of this paper is particularly apt to the Section T2 - Processing & Curation (https://www.ieee-ch.org)

**Reviewer #2: **This article investigates data management practices connected to a group of interdisciplinary physical collections to better understand how cyberinfrastructure tools may promote data reuse. They key insights cover the importance of metadata, the need for greater standardization of data, and a shortage of resources needed to upskill workers or hire staff with cyberinfrastructure skills.

The article would appeal to a moderately broad readership within the fields of museum and library studies. The study is well-designed and emphasizes the integration of insights from a diverse group of collection managers across the natural, physical, and social sciences and humanities. Yet the study also has limitations, which are not adequately addressed beyond a brief mention in the discussion on p.20. The qualitative insights derived from open-ended questioning of physical collection managers are interesting, but also subjective. This makes it somewhat challenging for a reader to integrate findings from this study into their own data management practices. A straightforward discussion of the benefits and limitations of the study design would go a long way toward addressing this issue.

The research design relies on the data curation profiling (DCP) method to guide the focus group questioning. The method is not fully explained in this article, although the questions asked to participants are listed, which is helpful. The article does reference other published work about the method (p.4), however a synopsis would be useful.

The article addresses a significant problem, distilling “the most important reuse facets that could inform the design of the cyberinfrastructure tools and services to improve reusability of all physical collections and resulting data” (p.2). It is increasingly important to curate data in ways that adhere to FAIR and CARE principles – and this article collects data that inform these forward-looking practices. The article also does a good job of referencing previous related research, including RDM (Research Data Management), responsible conduct in research, and the pressure for reproducibility in research (p.3).

I applaud the authors for making their data used in this study fully available without restriction, which is a testament to their commitment to open data and data reuse.

Overall, the paper is written in clear and correct English. There is one sentence I found confusing in the abstract (p.1), that I would recommend revising for clarity. The issue is the distinction between physical collections and interdisciplinary physical collections. The sentence reads, “Results indicated that physical collections attempt to use universal metadata and data storage standards to increase discoverability, but interdisciplinary physical collections and derived data reuse require more investments to increase reusability of these invaluable items.”

Overall, this is a well-researched and well-written paper that summarizes the results of important research. My main suggestion that would expand its relevance to a broader readership within museums and libraries is the addition of a discussion about the benefits and limitations of the study/RCP method.

6. PLOS authors have the option to publish the peer review history of their article (what does this mean?). If published, this will include your full peer review and any attached files.

Reviewer #1: **Yes: **Prof. Robin Boast

Reviewer #2: No

---

## [Author Response · Author response to Decision Letter 1]

18 Apr 2025

response to reviewers letter attached

---

## [Editor Report · Decision Letter 1]

30 May 2025

From Ice Cores to Dinosaurs: Physical Collections Managers’ Research Data Curation Perceptions and Behaviors

PONE-D-24-59078R1

Dear Dr. Bishop,

We’re pleased to inform you that your manuscript has been judged scientifically suitable for publication and will be formally accepted for publication once it meets all outstanding technical requirements.

Kind regards,

Peter F. Biehl, PhD

Academic Editor

PLOS ONE
---

## [Editor Report · Acceptance letter]

PONE-D-24-59078R1

PLOS ONE

Dear Dr. Bishop,

I'm pleased to inform you that your manuscript has been deemed suitable for publication in PLOS ONE. Congratulations! Your manuscript is now being handed over to our production team.

Kind regards,

on behalf of

Dr. Peter F. Biehl

Academic Editor

PLOS ONE